# Comprehensive Highlights of the Universal Efforts towards the Development of COVID-19 Vaccine

**DOI:** 10.3390/vaccines10101689

**Published:** 2022-10-10

**Authors:** Riyaz Ahamed Shaik, Mohammed Shakil Ahmad, Mansour Alzahrani, Nasser A. N. Alzerwi, Ahmad K. Alnemare, Musaed Reyzah, Haitham M. Albar, Salah Alshagrawi, Ahmed M. E. Elkhalifa, Raed Alzahrani, Yousef Alrohaimi, Turki M. Bin Mahfoz, Ritu Kumar Ahmad, Riyadh Ahmed Alahmdi, Nora Raid Saleem Al-baradie

**Affiliations:** 1Department of Family and Community Medicine, College of Medicine, Majmaah University, Al Majmaah 11952, Saudi Arabia; 2Department of Surgery, College of Medicine, Majmaah University, Ministry of Education, Al Majmaah 11952, Saudi Arabia; 3Otolaryngology Department, College of Medicine, Majmaah University, Al Majmaah 11952, Saudi Arabia; 4Department of Public Health, College of Health Sciences, Saudi Electronic University, Riyadh 11673, Saudi Arabia; 5Department of Haematology, Faculty of Medical Laboratory Sciences, University of El Imam El Mahdi, Kosti 1158, Sudan; 6Department of Basic Medical Sciences, College of Medicine, Majmaah University, Al Majmaah 11952, Saudi Arabia; 7Department of Pediatrics, College of Medicine, Majmaah University, Al Majmaah 11952, Saudi Arabia; 8Department of Otolaryngology, College of Medicine, Imam Mohammad Ibn Saud Islamic University (IMSIU), Riyadh 13317, Saudi Arabia; 9Applied Medical Sciences, Buraydah Private Colleges, Buraydah 51418, Saudi Arabia; 10Department of Family Medicine, King Abdullah Bin Abdulaziz University Hospital (KAAUH), Princess Nourah Bin Abdulrahman University, Riyadh 11671, Saudi Arabia; 11College of Medicine, Majmaah University, Al Majmaah 11952, Saudi Arabia

**Keywords:** coronavirus, COVID-19, vaccine, development, clinical trials

## Abstract

The world has taken proactive measures to combat the pandemic since the coronavirus disease 2019 (COVID-19) outbreak, which was caused by severe acute respiratory syndrome coronavirus-2 (SARS-CoV-2). These measures range from increasing the production of personal protective equipment (PPE) and highlighting the value of social distancing to the emergency use authorization (EUA) of therapeutic drugs or antibodies and their appropriate use; nonetheless, the disease is still spreading quickly and is ruining people’s social lives, the economy, and public health. As a result, effective vaccines are critical for bringing the pandemic to an end and restoring normalcy in society. Several potential COVID-19 vaccines are now being researched, developed, tested, and reviewed. Since the end of June 2022, several vaccines have been provisionally approved, whereas others are about to be approved. In the upcoming years, a large number of new medications that are presently undergoing clinical testing are anticipated to hit the market. To illustrate the advantages and disadvantages of their technique, to emphasize the additives and delivery methods used in their creation, and to project potential future growth, this study explores these vaccines and the related research endeavors, including conventional and prospective approaches.

## 1. Introduction

The SARS-CoV-2 virus is the cause of the infectious illness known as coronavirus disease (COVID-19). The majority of virus-infected individuals will contract a mild to severe respiratory disease and will recover without the need for special care. However, some people will move on to serious illnesses and will need to see a doctor. Serious sickness is more likely to strike older persons and those with underlying medical conditions, including cancer, diabetes, cardiovascular disease, or chronic respiratory diseases. COVID-19 can cause anyone to get very ill or to pass away at any age. Knowledge about the illness and the virus’s transmission method is key to preventing and reducing outbreaks. By keeping a distance of at least one meter between people, using a mask that fits properly, and frequently washing your hands or using an alcohol-based sanitizer, you may prevent infection in both you and other people. When your turn comes to be vaccinated, receive your vaccination and abide by any local advice. Note that when an infected person coughs, sneezes, speaks, sings, or breathes, the virus can spread from their mouth or nose in minute liquid particles. From larger respiratory droplets to tiny aerosols, these particles are diverse [1].

Increasing the manufacture of personal protective equipment (PPE), emphasizing the value of isolation after infection, obtaining an emergency use authorization (EUA) for therapeutic drugs or antibodies, and using well-known steroidal components are just a few examples. The world has taken significant steps to combat the COVID-19 pandemic since its outbreak due to severe acute respiratory syndrome coronavirus-2 (SARS-CoV-2). However, the disease continues to spread at an alarming rate, causing havoc for the health care system, society, and the economy. To stop this pandemic and allow society to return to normal, effective vaccines are urgently needed. A staggering amount of work has gone into researching, developing, testing, and evaluating COVID-19 vaccine candidates. By February 2021, some vaccines had been granted conditional approval, while others were very close to receiving the same. An increasing number of medicines undergoing clinical testing are projected to enter the market soon. One of the modern era’s most significant medical advances is universally acknowledged to be vaccination. One of the clearest instances of how this approach has been successful is the effective eradication of smallpox via vaccination programs, which may be used to eliminate a deadly disease and save countless lives. As a result of extensive immunization, the incidence of many childhood diseases has decreased, including polio and measles. Getting vaccinated every year against the flu is now generally considered the most effective means of defense against the seasonal variety of influenza. However, it usually takes at least 15 years from the time a new vaccine is proposed before it is tested in humans in a clinical trial when using the conventional vaccine development method [2]. 

To effectively shield a large population of otherwise healthy people, vaccines must evolve into increasingly complex biological products. As part of the long and eventful history of successful vaccine development against various pathogens, such as rubella, tuberculosis, diphtheria, tetanus, plague, influenza, measles, rotavirus, mumps, meningitis, rabies, polio, hepatitis, chickenpox, yellow fever, smallpox, malaria, tularemia, cholera, encephalitis, and pertussis, we are now moving one step further in the development of vaccinations against COVID-19, with the hope of achieving complete eradication [3].

Due to the extensive investigation and monitoring needed to ensure a vaccine’s security before it can be released, vaccine development and evaluation processes take a great deal of time. Vaccine drug studies are notoriously costly since they typically require recruiting a diverse group of people of varying ages, ethnicities, and health statuses to test the efficacy of the drug. To demonstrate the vaccine’s effectiveness and rule out any uncommon adverse effects, long-term surveillance is also needed. An unprecedented number of immunotherapies against COVID-19 have reached clinical trials and have been provisionally approved in just ten months since the outbreak began. This amazing speed was made possible by the availability of cutting-edge vaccination technology, the timely publication of the viral genome sequence, enormous/urgent market demand, and active collaboration among scientists worldwide, with an abundance of funding from a number of sources. Since the beginning of the epidemic, numerous organizations across the globe have committed substantial resources to the development of a vaccine against COVID-19 [4].

Recently, available vaccines (live attenuated vaccines, inactivated vaccines, mRNA vaccines, recombinant bacterial vector vaccines, recombinant subunit vaccines, recombinant viral vectors vaccines, DNA vaccines, virus-like particle vaccines, and synthetic peptides vaccines) were developed and evaluated quickly; however, they were found to be just as safe as earlier vaccines after a rigorous clinical data review by government regulators and assessment and rating groups, such as the United States Vaccines and Related Biological Products Advisory Committee (VRBPAC). The SARS-CoV-2 spike protein (S protein), or a fragment thereof, has been used as the immunogen in the development of multiple COVID-19 vaccine candidates. Viruses are able to enter human cells due to the S protein (a surface protein) that connects to the protein receptor, angiotensin-converting enzyme-2 (ACE2). Totaling 1273 amino acids, the S protein is split into two main sections, called the S1 and S2 domains. The S1-domain, which contains the receptor-binding region (RBD), is necessary for the S-protein to interact with the ACE2 receptor. The S protein was chosen as the immunogen because vaccines that are capable of inducing strong antibody responses will frequently have a significant influence on preventing viral entry into host cells during natural infection [5]. 

Research demonstrating that SARS-CoV-2-neutralizing antibodies collected from COVID-19 survivors were specific to the S protein or its RBD motif provided evidence in favor of this theory. The COVID-19 vaccine was developed using the knowledge gained from studies of SARS and MERS coronaviruses, including the identification of two proline modifications on the S-protein that preserve the protein’s antigenically optimal prefusion conformation. Animal models for assessing the efficacy of vaccines have been made possible, in part, by previous work conducted with the SARS and MERS vaccines. With the help of messenger RNA (mRNA) and vector vaccination methods, researchers were able to accurately build the antigen without undergoing the time-consuming and potentially dangerous protein purification procedures that are often required after a viral development phase [6].

This review first discusses the context of vaccine development and then examines the pros and cons of several vaccine delivery technologies. This review also focuses on nanotechnology in vaccine delivery systems and vaccination adjuvants, both of which are in their infancy. Anybody interested in learning more about COVID-19 vaccinations and the science behind them, as well as gaining a sense of the global effort necessary to develop such a vaccine, will find this review article to be a valuable resource.

## 2. The COVID-19 Vaccine Scenario

### 2.1. The Race for Development

Since the first reports of the COVID-19 pandemic emerged, a worldwide race has begun to develop effective vaccines. A review of vaccine research statistics by the World Health Organization (WHO) indicates that by February 2021, more than 40 countries and regions would be working on the COVID-19 vaccine, with the populations of at least half of these countries having one or more vaccinations in clinical trials. When compared with other countries and regions, the United States has significantly more vaccine candidates at both the clinical and preclinical stages. China ranks second, with active clinical testing in preclinical studies at present of from 10 to 13 more candidates. Third place goes to Canada, where twelve potential treatments are still in the preclinical stages and five are currently undergoing human testing. Two vaccines were developed in the United States of America (Johnson & Johnson and Moderna); one was developed in a collaboration between Germany (BioNTech) and the United States of America (Pfizer); one was developed in the United Kingdom (AstraZeneca); four vaccines were developed in China; one was developed in Russia (granted conditional permission) encouraging one or more nations to move forward with clinical trials [7,8].

### 2.2. Clinical Trial Reports

There were 256 vaccine candidates developed by the end of February 2021, with 182 still in the preclinical stages and 74 undergoing clinical trials. Sixteen candidates out of 74 candidates are participating in large-scale phase-3 or phase-4 clinical investigations to further validate vaccine candidate safety and effectiveness. Several methods were used to create these vaccine candidates. These included recombinant protein-based vaccines (virus-like particle (VLP) vaccines and protein subunit vaccines), nucleic acid vaccines (mRNA-based vaccines and DNA-based vaccines), whole virus vaccines (inactivated), and viral vector vaccines. At present, there are 96 different candidates, with 24 currently undergoing clinical trials and another 68 still in the early stages of development. Non-replicating viral vectors make up 13.3%, while replicating viral vectors comprise 10.2%, inactivated vaccines make up 9.8%, and mRNA and DNA candidates each contribute 12.1%. Vaccines are currently in the fourth and final phase of clinical testing in the US, China, the UK, and Germany (post-market studies). India is now the only nation developing a live-attenuated vaccine. A potential inactivated COVID-19 vaccine solution has candidates in China, India, and the UK. All seven nations, with the possible exception of Germany, are working on vaccines based on proteins. Even though the United States is not generating any live attenuated vaccine, it has more vaccines in clinical testing than any other country. China is a close second. Vaccines against COVID-19, made using either viral vectors or mRNA, are undergoing extensive clinical trials in both Germany and the UK. Currently, in the early stages of clinical testing, vaccinations are based on protein subunits, viral vectors, and DNA from South Korea and Canada [9,10,11].

### 2.3. Computational Roles in Vaccine Development

A field of biology called bioinformatics is concerned with using computer-based techniques to examine biological systems and make exact predictions that may come to pass in laboratory experiments and clinical trials. The life sciences have been transformed by the development of computer-based biological procedures, and medical facilities are no longer under as much financial strain due to the reduction in the high expenses of animal sacrifice and laboratory labor. When we need to create servers for sorting these molecules using machine learning (ML) approaches, in silico techniques are beneficial for classifying proteins based on their structure and function. In the discovery and development of COVID-19 vaccines, the following are the most prominent aspects that were taken into consideration: virtual screening of potential drugs against this pathogen, prediction of SARS-CoV2 structures using the in silico method, the application of anti-microbial peptides and peptidomimetics against COVID-19, the prediction of potential natural compounds to suppress SARS-CoV2, for designing vaccines, and artificial intelligence (AI) and ML methods [12,13].

## 3. Vaccines

Several platforms have been explored for the development of vaccines, each with a unique blend of benefits and drawbacks. To combat this pandemic effectively, the entire globe will most likely need a number of different licensed vaccines to meet the requirements of sufficient production volumes and broad population coverage, storage, and transit requirements, while maintaining a satisfactory vaccine safety profile. The information covered in this section was drawn from the following cumulative and overlapping reports [14,15,16,17,18,19,20,21,22,23,24,25,26,27,28,29,30,31,32,33,34,35,36,37].

### 3.1. Conventional Whole-Virus Vaccines

Smallpox, BCG, and measles immunizations have all used conventional whole-virus vaccines for decades. These vaccines often include both the live attenuated and inactivated strains of the virus. To protect against COVID-19, vaccinations are administered using an attenuated or inactivated form of the SARS-CoV-2 virus. It is anticipated that immune responses will target not only the SARS-CoV-2 protein but also other virus particles. Due to the lengthy process of virus cultivation and the necessity for specialized biosafety level-3 manufacturing procedures, these COVID-19 vaccines are far more challenging to produce.

#### 3.1.1. Live-Attenuated Vaccines

Historically, disease-causing viruses have been serially passed through cultured cells to create live-attenuated vaccines, with the goal of reducing their ability to replicate and, consequently, their pathogenicity. Vaccines that use attenuated viruses still work, as their ability to replicate is not affected by this reduction in their pathogenicity. A long-lasting cell-mediated immune response and antibody-mediated (humoral) response can be induced by these vaccines because they mimic the effects of actual infection; however, they must be thoroughly tested for safety. Due to the potential for infection from the vaccine’s live pathogens, live attenuated viral vaccines are rarely used in immunocompromised patients, although they provide no danger of spreading the illness.

##### New Vaccine

By deoptimizing the codons, Codagenix and the Serum Institute of India were able to create a live-attenuated COVID-19 candidate. Trials with a single intranasal dose have begun in Phase 1. The process of preclinical testing has begun for an additional four candidates. However, it is important to note that some people are worried that immunizations of this kind could actually cause a resurgence in virulence under specific circumstances.

#### 3.1.2. Inactivated Vaccines

To create inactivated vaccines, the viruses are grown in a growing medium and then neutralized by chemicals, heat, or radiation to produce inactivated vaccines, also known as killed immunizations. Vaccines for certain diseases, including hepatitis A, polio, and measles fall under this heading. These vaccines use attenuated viruses that cannot replicate and, hence, cannot cause disease in immunocompromised people. The immune response that they elicit is largely antibody-mediated rather than cell-mediated, making them less effective than live-attenuated vaccines and requiring booster shots to provide full protection.

##### Clinical Trials

The inactivated COVID-19 vaccines have undergone preclinical testing (11 candidates out of 21 candidates); ten have come into clinical trials, and five are in Phase-3 trials.

##### CoronaVac

Sinovac developed CoronaVac with adjuvant (aluminum hydroxide) and β-propiolactone as an inactivating agent. A phase-1/2 clinical trial showed promising tolerability and immunological evidence, while a phase-3 trial showed an overall efficacy of 50.7%; China Inspection Body and Laboratory Mandatory Approval (CMA) was granted in China.

##### BBIBP-CorV

After promising results in Phase-2 investigations, Sinopharm is moving forward with Phase-3 trials of BBIBP-CorV. Even though its claimed efficacy is just 79.3 percent, the CMA in China has still sanctioned its use.

##### BBV152

Recently approved in India, BBV152 from Bharat Biotech is a Phase-3 clinical study participant.

### 3.2. Recombinant Viral Protein-Based Vaccines

Even though producing whole viral vaccines is based on a tried-and-true method, strict protocols and regulations must be adhered to. Vaccines based on viral proteins are produced using recombinant techniques; these vaccines contain no genetic material and only the viral proteins exclusively (immunogens), making them both more secure and less likely to result in infection because no complete virus is utilized.

#### 3.2.1. Protein Subunit Vaccines

Protein subunit-based vaccine-oriented immunizations use viral proteins or viral protein fragments as the antigen. Numerous licensed vaccines, such as those used against HBV and DTP, take advantage of this effective approach. Due to their intrinsic safety and the fact that they can be employed using specific efficient protein purification techniques, these vaccines make up the bulk of vaccine progress. The S-protein or a portion of it is used as the basis for most of these vaccinations (RBD). Given that the S-protein is membrane-bound and is composed of several subunits, it may be challenging to express and produce the whole S protein. The RBD fragment, which is similar to the full-length S-protein in its ability to elicit powerful neutralizing antibodies, is considerably smaller and easier to make. It does, however, leave out some other rather crucial epitopes that the full-length S-protein provides. Therefore, vaccines developed using RBDs may be less effective than those using S-proteins. Protein subunit vaccines, such as inactivated vaccines, primarily stimulate antibody-mediated immunity with some T-cell response activation. Vaccinations of this type typically necessitate the use of adjuvants to effectively trigger an immune response and maximize the vaccine’s efficacy.

##### Clinical Trials Report

Twenty-four subunit vaccines, such as those developed by Anhui Zhifei Longcom Biopharmaceutical, Novavax, Sanofi/GlaxoSmithKline (GSK), and Kentucky Bioprocessing, have reached clinical trials, while 68 more vaccine possibilities are undergoing preclinical evaluation.

##### NVX-CoV2373

Novavax developed NVX-CoV2373, a nanoparticle-based S-protein recombinant vaccine with the Matrix M1 adjuvant containing saponin. In an interim review of a phase 3 study, it demonstrated an efficacy rate of 89.3%, combined with immunological data and reasonable safety, as shown in earlier clinical trials. The Novavax UK study revealed that phase-3 research demonstrated a 96.4 percent efficiency in contrast to the first viral strain and offered 86.3 percent efficiency alongside the B.1.1.7 variant, for an impressive 89.7 percent overall. In particular, none of the patients treated with NVX-CoV2373 developed a serious disease.

##### UB-612

COVAXX’s UB-612 vaccine, a peptide-based vaccine, promotes cytotoxic T-cell-mediated immunity and B-cell-mediated immunity by targeting the RBD as well as epitopes on nucleocapsid proteins and the viral structural membrane.

##### Cuban Abdala

The Genetic Engineering and Biotechnology Center (CIGB) created the recombinant protein subunit vaccination known as Abdala, using the yeast *Pichia pastoris* as the microorganism for expression. The recombinant SARS-CoV-2 protein receptor binding domain (RBD) and aluminum hydroxide gel are both components of its composition. The entire immunization regimen for Abdala involves three 0.5 mL doses, given intramuscularly (IM) in the deltoid area on days 0, 14, and 28. It can be mixed with other vaccinations or used as a booster. Abdala was well tolerated in the clinical studies (RPCEC00000346, RPCEC00000359, and RPCEC00000363) that evaluated its safety, immunogenicity, and effectiveness. No serious side events were identified. The Abdala vaccine was safe, well tolerated, and elicited humoral immune responses against SARS-CoV-2 with a 50 mg dosage, applied in a 0–14–28 days schedule in a Phase-1/2, randomized, double-blind, placebo-controlled experiment conducted in “Saturnino Lora” Hospital, Santiago de Cuba. Abdala demonstrated an effectiveness of 92.3% (95% CI: 85,795.8) in a multi-center, randomized, double-blind, placebo-controlled Phase-3 research project including 48,290 adult volunteers. However, no peer-reviewed article on Abdala’s phase-3 randomized clinical trials has been published to date.

#### 3.2.2. Virus-Like Particle (VLP) Vaccines

A subset of protein subunit vaccinations, VLP vaccines are an evolution of protein subunit immunization. These particles are composed of viral capsid/coat proteins, which self-assemble into a structure resembling a capsid, without the viral genome or other nonstructural virus proteins, when recombinantly created from a host organism. However, these noncontagious particles provide a framework for chemically linking or arranging several copies of the antigen (or epitope). Epitope/antigen clustering over the surface of VLPs promotes antibody reactivity and cognate B-cell activation. One of the two VLP vaccines under clinical trials is an AS03-adjuvanted vaccine created by Medicago Inc., while the other is an AS03-adjuvanted vaccine developed by SpyBiotech/Serum Institute of India.

### 3.3. Viral Vector Vaccines

Antigen-coding DNA fragments are delivered to host cells, via nonpathogenic viral vectors in vector vaccines, to confirm the antigen through the protein-making machinery of the host cells. In comparison, the conventional entire virus and protein-based vaccinations introduce antigenic proteins directly into host cells to elicit immunity. The size of the antigen gene to be inserted, measured in base pairs, first determines the best viral vector to use because some vectors are noticeably more sensitive than others. Additionally, there are challenges related to manufacturing, the genetic integrity of the vector, and the sample population’s tolerance and preexisting resistance.

#### 3.3.1. Types of Viral Vector Vaccines

In general, these can be divided into two categories: those that do not multiply viral vectors and those that do. Vaccines given through viral vectors have a far higher chance of triggering indigenous antibody synthesis, which, in turn, triggers both humoral immunity and innate immunity. Viral vector-based vaccinations do not need to be transported or stored at incredibly low temperatures, and they are easy to design and produce on a large scale. However, preexisting vector sensitivity may limit the vector’s ability to transmit hereditary data to the host organism, which would lessen the efficiency of the vaccination in the long term.

##### Nonreplicating Viral Vector Vaccines

Prior to the pandemic, no vaccines containing nonreplicating viral vectors had been approved for use. However, research is currently underway to evaluate 12 vaccine candidates against COVID-19. Many of them are based on adenoviral (Ad) vectors that are incapable of replicating within the human body. The majority of the time, this incapacity is accomplished by deleting a gene in the viral vector that produces a structural protein, which stops an infected cell from producing virion. A transgenic host cell and helper virus must supply the crucial structural component to build the vaccine vector.

##### Ad5 nCoV

Based on human adenovirus 5 (Ad5), CanSino Biologics’ vaccine has a 1-month overall effectiveness of >60% after a single injection inoculation.

##### JNJ-78436735

Four weeks after vaccination, the human adenovirus 26-based single-dose vaccine JNJ-78436735 had a 66% efficacy rate for preventing mild to severe COVID-19 infection.

##### AZD1222

AZD1222 was created by AstraZeneca and Oxford University. It is based on an adenovirus from chimpanzees against which humans are unlikely to have preexisting immunity. The safety and effectiveness of the vaccine were validated by the interim analysis of a phase 2/3 study (70.4%). Each of these vaccinations against vector-borne viruses has received approval in at least one country.

##### VXA-CoV-2-1

The adenovirus 5 oral vector vaccine VXA-CoV-2-1, created by Vaxart, is now undergoing phase-1 studies. Prime-boost immunization (heterologous) with diverse vectors may lessen the effect of adenoviral vectors, which have the ability to stimulate the immune function against the vector and reduce antigen-induced activity.

##### Sputnik V

The Gamaleya Research Institute has developed two varieties of Sputnik V; the first was based on the adenovirus-5 vector, and the second was based on the adenovirus-26 vector. These two formulations are administered successively over the course of three weeks. Its 91.6% effectiveness against COVID-19 was revealed by the intermediate findings of the Russian Phase-3 clinical trial. This product is offered in both chilled (−18 °C storage temperature) and cryo-frozen (−28 °C storage temperature) formulations.

##### Replicating Viral Vector Vaccines

Replicating viral vectors can reproduce in infected cells, allowing for a lower dosage to induce resistance compared to nonreplicating viral vector vaccines. Replicating viral vector vaccine pathogenicity or longevity might raise safety concerns, especially for immunocompromised people.

##### Clinical Trials

The Jiangsu Provincial Center for Disease Prevention and Control developed an influenza-based RBD replicating the viral vector intranasal vaccine, as one of six options undergoing clinical testing. The recombinant vesicular stomatitis virus (rVSV) was used in the creation of the licensed Ebola vaccine, ERVEBO, and was also used by the Israel Institute for Biological Research and the Weizmann Institute of Science to create VSV-S. Patients with wild-type VSV infections usually exhibit no symptoms or only very mild flu-like indications.

### 3.4. Nucleic Acid Vaccines

Similar to vaccines made with viral vectors, nucleic acid vaccines work by introducing genetic material into host cells, which then employ the machinery for protein synthesis to create immunogens. The in situ generation of these exogenous immunogens inside the human host substantially enhances T-cell induction and antibody generation, as shown in patients who have recovered. Additionally, nucleic acid vaccines may be created on a larger scale.

#### 3.4.1. DNA Vaccines

Plasmid DNA with a transgene expressing the antigen, as well as a protein mammalian expression promoter—in the case of vaccines, the S-protein—is used to make vaccines based on DNA. They are incredibly stable at room temperature and are easy to generate in vitro, making it possible to store and distribute them easily. Because of this advantage, they could be especially well suited for use in the endemic regions of underdeveloped countries. Additionally, the platform is easily adaptable for providing a vaccine for a new antigen.

##### Clinical Trials

In clinical research, DNA vaccines delivered via traditional needle administration, without adjuvants or other delivery agents, were associated with low immunogenicity. This issue has been addressed by elevating the immunogenicity of DNA vaccines during immunization, using physical delivery techniques, such as electroporators and gene guns. Despite the fact that there is as yet no approved DNA vaccine, 11 candidates are now in the clinical stage of development.

##### INO-4800

INO-4800, an intradermally administered DNA vaccine created by Inovio Pharmaceuticals, generates the entire S-protein and is now undergoing a phase 2/3 trial. INO-4800 is generally well tolerated in volunteers, according to the initial results from a phase-1 clinical study. Additionally, it is immunogenic in all individuals, eliciting neutralizing T-cell reactions or antibodies.

##### AG0301 and AG0302

Presently, AnGes/Osaka University’s AG0301-COVID-19 and AG0302-COVID-19, as well as Cadila Healthcare Limited’s ZyCoV-D, are in the late stages of clinical studies.

#### 3.4.2. mRNA Vaccines

The genetic code for protein antigens is included in mRNA that has been encapsulated as a vaccination (LNPs). Following vaccination, the LNP-mRNA is transported to the host tissue cytosol, where it acts as a blueprint for the creation of protein antigens. Through the host cell’s cytosolic protein synthesis process, various COVID-19 mRNA vaccination candidates transmit and translate the genetic code of the full-length S-protein. As with vector and DNA vaccinations, mRNA vaccines may promote both T-cell response and antibody formation via the production of protein antigens in the injected individual’s cells, similar to vector and DNA vaccines. In addition, antigen synthesis is transient following mRNA immunization, limiting its systemic retention. These qualities indicate that mRNA may be an option for rapid, inexpensive, and efficient vaccine manufacturing.

##### Clinical Trials

Approved

Several nations have obtained conditional approval for two COVID-19 mRNA vaccines developed by Moderna (mRNA-1273) and Pfizer/BioNTech (BNT162b2). Six other vaccines are undergoing clinical testing. Less than two months after the viral genome sequence release, Moderna developed mRNA-1273 for clinical testing. After the Phase-1 trial finished in July 2020, the findings attested to the drug’s safety and protective immune reaction. Similarly, Pfizer/BioNTech verified the phase-1 safety and immunogenicity data for BNT162b2 and advanced the vaccine to phase-2/phase-3 trials. Good phase-3 safety and effectiveness findings for their mRNA vaccines were disclosed by Moderna and Pfizer/BioNTech in November (94.1% and 95%, respectively). In addition, the vaccine developed by Moderna was 100% effective against severe COVID-19 infection. Due to the fragility of RNA molecules, the storage and transit of particular mRNA candidates require intensive monitoring of the cold chain. Moderna and Pfizer/BioNTech mRNA vaccines must be stored at 2 °C and 8 °C, respectively (Table 1).

Other Vaccines

The German company CureVac has said that their phase-2 testing mRNA vaccine candidate CVnCOV is stable for three months at 5 °C. Additionally, Arcturus Therapeutics has created ARCT-021, a single-dose, freeze-dried, identity mRNA vaccine that may be kept at room temperature.

### 3.5. Promising Vaccine Candidates

Surprisingly, as vaccination platforms, these vaccines only use mRNA, nonreplicating viral vectors, inactivated vaccines, and protein subunits (Figure 1). All these medications must be administered intramuscularly. While the bulk of the vaccines described require two doses, Johnson & Johnson and CanSino Biologics’ viral vector-based vaccines require only one. Despite being the first to be authorized, mRNA vaccines cannot be disseminated in remote or underdeveloped locations because of their strict temperature storage requirements. The temperature requirements for all the other vaccine types are quite flexible, making them simpler to administer in less-developed regions.

Live attenuated vaccines (Covivac, Covaxin, and QazVac), inactivated vaccines (CoronaVac, BBIBP-CorV, and BBV152), mRNA vaccines (BNT162), recombinant bacterial vector vaccines (Symvivo), recombinant sub-unit vaccines (ZF2001, MVC-COV1901, CIGB-66, and NVX-CoV2373), recombinant viral vector vaccines (Sputnik V, Sputnik Light, Covishield, Ad26.COV2.S, and Ad5-nCoV), DNA vaccines (ZyCOV-D), virus-like particle vaccines (CoVLP), and synthetic peptide vaccines (EpiVacCorona).

### 3.6. Vaccines Used under Emergency Authorization

Nearly 190 countries have approved COVID-19 vaccines under an emergency use authorization (EUA). Some of the vaccines, such as Moderna, Pfizer/BioNTech, CanSino, Oxford/AstraZeneca, Sinopharm, EpiVac, and Sinovac, have received full authorization in specific countries. However, only seven vaccines, namely, Pfizer/BioNTech (BNT162b2), Moderna (mRNA-1273), Oxford/AstraZeneca (AZD1222), Janssen (Ad26.COV2.), Sinopharm (BBIBP-CorV), Serum Institute of India—Covishield (Oxford/AstraZeneca formulation), and Sinovac (CoronaVac) have received a WHO Emergency Use Listing (EUL).

#### 3.6.1. Sinopharm

China was the country in which emergency usage of Sinopharm for those aged 18 to 59 was first authorized. The vaccination doses need to be stored at between 2 and 8 degrees Celsius. It is given on day 0 and as a booster on day 21. It has been demonstrated to have 95–100% seroconversion. However, nausea, local responses, and fever were among the unpleasant side effects that were recorded.

#### 3.6.2. Sinovac

China was the country in which emergency usage of Sinovac for those aged 18 to 59 was first authorized. The vaccination doses must be stored at between 2 and 8 degrees Celsius. Both the first dosage and the booster dose are administered on day 0 and given as a booster on day 28. Seroconversion rates range from 97 to 100%. Only a few patients have reported mild local effects.

#### 3.6.3. Covaxin

Emergency use authorization was granted for the fully inactivated vaccine, created by Bharat Biotech (India) in the clinical trial phase. The safety and immunogenicity statistics were satisfactory.

#### 3.6.4. CoviVac

Another inactivated (whole virion) vaccine, CoviVac, which was created by the Chumakov Center, is authorized in Russia and is provided in dual doses with a 2-week gap. It was rolled out and stored at standard refrigerator settings, ranging from 2 to 8 degrees Celsius. However, due to differences in age groups, geographical regions, settings, and variations, these vaccines are not comparable. The vaccine is currently in phase-3 clinical testing.

#### 3.6.5. QazVac

QazCovid-in (QazVac) was developed by a biological safety research institute. It has been shown to be well-tolerated and to produce humoral immunity that lasts for six months. According to the phase-1 experiment, 59% of patients developed neutralizing antibodies after one treatment, and 100% developed them after two. In a phase-2 study, subjects between the ages of 18 and 49 had 100% seroconversion, even after one dosage, while those between the ages of 50 and above demonstrated 92–94% seroconversion following one dose and 100% seroconversion following two doses on days 0 and 21. Injection site soreness, fever, weakness, malaise, and tiredness were all recorded as side effects.

#### 3.6.6. COVIranBarekat

The early stages of clinical studies for the inactivated whole viral vaccination COVIranBarekat showed immunogenicity in up to 93% of participants. Phase 3 of the study is still in progress after receiving EUA.

#### 3.6.7. KCONVAC

Minhai Biotechnology and Shenzhen Kangtai Biological Products collaborated to produce KCONVAC. It is an authorized inactivated vaccine that is manufactured from Vero cells for use in emergencies in China.

#### 3.6.8. Sputnik V

The vaccine is available to all ages between 18 and 60. The immunization needs to be stored between 2 and 8 degrees Celsius. Day 0 (rAd26-S) and day 21 (the booster dosage) are when it is administered. (rAd5-S). It supposedly has a 100% seroconversion rate. Local responses and moderate fever are side effects.

##### Clinical Trials

This recombinant adenovirus-based vaccine (heterologous), named Sputnik-V or Gam-COVID-Vac, demonstrated 91.6% effectiveness in the preliminary findings from a previous study (Phase 3). The primary outcome analysis of the research comprised 19,866 individuals. COVID-19 positivity was discovered in 0.1% of 14,964 vaccination group participants and 1.3% of 4902 placebo group participants when the second vaccination dosage was administered (day 21), with a vaccine efficacy of 91.6%. A total of 21,977 people were examined for safety analysis, including 16,501 examined in the vaccine arm and 5476 in the placebo arm. There were <0.5% of serious adverse reactions in both arms, although none were thought to be related to the vaccination. Three fatalities that were determined to be unrelated to immunization occurred in the vaccine arm group, compared to a single fatality in the placebo arm group.

#### 3.6.9. Oxford/AstraZeneca

Early clinical studies of viral vector-based vaccination revealed a positive humoral and cellular response. The safety results were also satisfactory, but acetaminophen was employed as a prophylactic therapy to prevent unpleasant side events such as chills, myalgia, and fever. After a single dose of vaccine, the immune response continued for up to 8 weeks. Additionally, it was shown that immunization was acceptable and comparable immunogenically in older people (>70 years). Additionally, it has been shown that patients who have strong antibody responses tolerate a booster dosage better. The two-dose regimen, on the other hand, demonstrated the lowest vaccination effectiveness of 10.4% for the B.1.351 variation.

##### Clinical Trials

Four phase-3 blinded, randomized controlled trials of the hAdOx1 nCoV-19 vaccine (AZD1222) have been conducted in South Africa, the UK, and Brazil. Early analyses of these pooled data have been made public. With a subgroup of participants recruited in the UK, receiving a half-dosage as the first dose (LD) and the standard dose as a second dose in the LD/SD cohort, ChAdOx1 nCoV-19 is evaluated in two standard doses (5X 1010 virus particles each) in the SD/SD cohort. Symptomatic COVID-19, confirmed by a nucleic acid amplification swab test (NAAT) 14 days after the second dose was the major efficacy result. Data from an almost 6-month period revealed an enrollment of 23,848 individuals. The primary efficacy endpoint was achieved by 27 of 4440 participants (0.6%) in the ChAdOx1 nCoV-19 vaccination group and 71 of 4455 (1.6%) in the control group in the SD/SD cohort. When the effectiveness of the vaccination was calculated as 1-relative risk using an age-adjusted Poisson regression model, it was found to be 62.1%. In the LD/SD cohort, three of 1367 (0.2%) in the vaccination group and 30 of 1374 (2.2%) in the control group met the primary efficacy end objective. The vaccination’s efficacy was estimated to be 90.0%. The estimated percentage of vaccinations that were successful overall was 70.4% (95% confidence interval, 54.8–80.6%). According to the safety analysis, there were 84 occurrences in the vaccination group and 91 events in the control group over the median follow-up of 3.4 months. Voysey et al. also discovered that participants with more than six weeks between doses (65.4%) showed higher vaccination efficacy than those with fewer than six weeks between doses (53.4%). According to a recent study, older volunteers who received a single dose of either the BNT162b2 or ChAdOx1 COVID-19 vaccination were less likely to develop symptoms or develop a severe COVID-19 infection.

#### 3.6.10. CanSino

Anyone above the age of 18 may use this vaccination in China, according to a license. This vaccination should be stored at temperatures ranging from 2 to 8 degrees Celsius. On day 0 and day 28, respectively, the first and second doses are administered. It is believed to have a seroconversion rate of 59–61%. Injection site discomfort and fever are common side effects.

#### 3.6.11. Pfizer/BioNTech

Kuwait, the United Kingdom, Saudi Arabia, the United States of America, Switzerland, the United Arab Emirates, Qatar, Canada, and Oman have all licensed this vaccine for those aged 18 to 55. The vaccination must be refrigerated between −8 °C and −6 °C. It is given on day 0, while the booster dose is given after 3 weeks. Local responses, lethargy, headaches, nausea, and other side effects are common. Antibodies were raised 1.8-fold to 2.8-fold in a restorative serum section and were 95% effective in averting COVID-19 infection. The Vaccine and Related Biological Products Advisory Committee (VRBPAC) has suggested booster doses exclusively for adults over the age of 65 and those at high risk of serious disease.

##### Clinical Trials

BNT 162b2 was tested in a worldwide placebo-controlled, observer-blind pivotal effectiveness trial. The existence of COVID-19 at least 7 days after the second treatment was the main outcome. According to a published study, 162 subjects in the placebo group and 8 participants who received the vaccination contracted COVID-19. As a result, the vaccination is said to be 95% effective. Injection site responses, tiredness, and headache were among the reported side effects. In their investigation, scientists evaluated vaccination effectiveness as 100X (1-IRR). The incidence rate ratio (IRR) contrasts the number of illnesses in the placebo group with the number of confirmed COVID-19 cases per 1000 person-years of follow-up in the immunization group. BNT 162b2 has also demonstrated efficacy in a large population in a mass-vaccination environment. Using the Kaplan—Meier estimator, vaccine effectiveness was computed as one minus the risk ratio. For documented infection, with symptomatic COVID-19, it was 57%; the estimated effectiveness was 46%; for severe disease, 62%; for hospitalization, 74%; and for preventing mortality, 72%. For documented infection in terms of symptomatic COVID-19, it was 94%; the effectiveness was 92%; for severe disease, it was 92%; and for hospitalization, it was 87%.

#### 3.6.12. Moderna

Individuals under the age of 18 are permitted to use the Moderna vaccine in the United States of America. It should be stored at 2–8 degrees Celsius. The booster dosage was administered on day 28, while the first dose was administered on day 0. It has been proven to have a 100% seroconversion rate. Local responses, fever, tiredness, and other common side effects were recorded. Substantial titers of binding and neutralizing antibodies were still present in mRNA-1273 at approximately 90 days after the second dose of immunization. Figure 2 depicts the progression of Moderna mRNA-1273 vaccine development.

##### Clinical Trials

After being injected into the bloodstream, the lipid nanoparticle-encapsulated mRNA-1273 vaccine is designed to express the comprehensive spike protein. The phase-3 randomized observer-blinded placebo-controlled research had 15,210 participants in both the vaccination and placebo arms. Following the second dosage, 14 days later, the incidence of symptomatic COVID-19 was >50 per 1000 individuals in the placebo arm and ~3.5 per 1000 individuals in the vaccination arm. The efficacy of the vaccination was found to be 94.1%. A percentage drop in the major endpoint’s hazard ratio served as the measure of efficacy. In the placebo group, it was experienced by thirty individuals, all of whom had significant symptoms. Systemic and local responses in the vaccination group were brief but common. ADRs such as erythema, headache, induration, fever, myalgia, pain at the site of injection, and arthralgia were reported. Serious adverse events occurred almost equally often in all groups.

#### 3.6.13. Ad26.COV2. S

The USFDA has authorized Ad26COVS1, also known as JNJ-78436735 or Ad26COVS1, as a single-dose COVID-19 vaccination.

#### 3.6.14. ZyCoV-D

The first DNA-based vaccination to be approved for use in humans is ZyCoV-D. The intradermal vaccination was found to offer 100% protection against mild sickness and 67% protection against COVID symptoms in individuals aged 12 and over. Based on clinical tests carried out during the second wave of the pandemic in 2021, the vaccine has been approved under EUA in India. Phase 1 research demonstrated tolerance and immunogenicity when three doses were given at 28-day intervals (days 0, 28, and 56). The PharmajetTropis^®^ needle-free injection and needle-based administration were both used in the research. The reported adverse effects were fever, tenderness at the injection site, and joint discomfort. After receiving a 2 mg needle-based or 2 mg needleless injection, 50% or 80% of patients, respectively, developed antibodies by day 84 according to the neutralization antibody titers, based on a live virus test.

#### 3.6.15. EpiVacCorona

The Russian authorities have granted licenses for the peptide-based vaccine, EpiVacCorona. According to reports, all subjects responded to the immunization immunologically in all cases. The Vektor State Research Center of Virology and Biotechnology is where EpiVacCorona was created. It is administered in two doses, separated by 3 weeks, and comprises the SARS CoV2 antigen, coupled with a protein carrier and an aluminum adjuvant. There have been no major safety concerns raised about the vaccination. Turkmenistan has granted full permission for its use.

#### 3.6.16. ZF2001

Anhui Zhifei Longcom has developed this protein subunit-based vaccination. The data from phases 1 and 2 imply tolerance and immunogenicity. After 14 days, the seroconversion rate of the neutralizing antibodies was 93–97%. In a phase-3 study, the ZF2001 vaccination was found to be effective against alpha-variant disease (81.7%) and delta variant sickness (77.5%). It is administered in three doses (days 0, 30, and 60) and has been demonstrated to maintain neutralizing efficacy in SARS CoV2’s delta version.

#### 3.6.17. CIGB-66

Cuba’s Center for Genetic Engineering and Biotechnology (CIGB) has produced CIGB-66, popularly known as Abdala, a protein subunit-based vaccination. In Cuba and Venezuela, CIGB-66 is permitted for emergency usage. Phases 1 and 2 of the clinical trials showed that the drug was tolerable and immunogenic, and the findings of the phase-3 investigation are awaited. The vaccine is administered intramuscularly on days 0, 14, and 28.

#### 3.6.18. MVC-COV1901

Medigen and Dynavax collaborated to create a protein subunit vaccine comprising the CpG1018 and S-2P antigens, adjuvanted with aluminum hydroxide. In Taiwan, the vaccine has been approved for emergency use. MVC-COV1901 has proven tolerance and immunogenicity in phase-1 and phase-2 investigations, with a seroconversion percentage of 99.8% following the second dose at 28 days.

## 4. Adjuvants

### 4.1. Role of Adjuvants

Reducing the quantity of antigen required to trigger a robust immunological response, producing a chemokine milieu and inflammatory cytokines, adjuvants aid in the prevention and treatment effectiveness of vaccines and divide the routes of type-2 vs. type-1 differentiation to polarize the helper T-cell response. They are mostly utilized in vaccinations designed to protect against protein subunits. Although the exact mechanisms underlying these events are unknown, it is widely believed that they entail increased antigen presentation, antigen accumulation, the activation of the immune system (innate) via Toll-like receptors, and immune cell recruitment, which act as an alarm system against pathogen attack. Adjuvants are a common component of vaccines, including those for COVID-19. Adjuvants are recognized and well-known chemicals [38].

### 4.2. Natural Products

In preclinical and experimental settings, numerous endogenous cytokines (e.g., interleukins, GM-CSF, interferons, etc.), natural compounds (e.g., β-galactosylceramide and polysaccharides), and synthetic chemicals were studied. As the mechanisms behind the activities of these adjuvants have been investigated, it has become clear that many of these medicines interact with the pattern recognition receptor (PRR) system of the adaptive immune system. Through the promoter of interferon genes protein (STING), inflammasome, PRR, Toll-like receptors (TLRs), etc., have been established to identify pathogen- and host-associated molecular patterns (PAMPs and DAMPs). Many of the aforementioned adjuvants have been demonstrated to elicit Toll-like receptor-mediated immune responses via the NLR family pyrin domain-containing protein 3 (NLRP3)-inflammasome. Molecules such as the imidazoquinoline derivatives have also been explored as potential vaccine adjuvants since they induce reactive immunological responses in this way [39].

### 4.3. Global Market

According to some estimates, in 2016, the market for vaccine adjuvants reached USD 300,400 million and is predicted to reach USD 1 billion by 2027. As a result, it is not unexpected that several pharmaceutical corporations and academic institutions have invested in this sector. The Toll-like receptor agonists 3, 4, 7, and 8 have piqued the interest of researchers, who have found that the cyclic nucleotide activation of STING could significantly improve the T-cell response/IgG response toward the spike protein of SARS-CoV-2, compared to the aluminum hydroxide adjuvant [40].

## 5. mRNA Sequence Modification

Many strategies have been devised to enhance the translational efficiency and stability of mRNA sequences while reducing their immunogenicity. The open reading frame (ORF), followed by the 5′ untranslated region and 3′ untranslated region (UTRs), 5′ cap, and 3′ polyadenylation (poly(A)) tail, are the basic building blocks of in vitro-generated mRNA vaccines, similar to human mRNA. For mRNA to be translated and remain stable on the ribosome, it needs a cap, a poly (A) tail, and some UTRs. Mammalian mRNA requires a functional 5′-cap structure, which is a 7-methylguanosine triphosphate link to the mRNA’s first nucleotide, for efficient translation. The translation of mRNA must first complete the 5′ capping process. The translational efficiency and stability of mRNA vaccines have been studied and analogs of the mRNA cap have been developed. The host immune system also identifies IVT mRNA without a cap as being foreign RNA through the Toll-like receptor-7 and Toll-like receptor-8. The immunogenicity of IVT mRNA could be reduced by capping it, as this would prevent the mRNA from being identified by the host immune system. For an mRNA to be stable and translated, its 3′ end must have a long poly (A) tail. UTR sequences are incorporated into mRNA to increase its stability and lengthen its half-life, which in turn increases the efficiency with which it is translated. The integration of human β-globin UTR into an mRNA’s 5′-UTR and 3′-UTR areas has indeed been proven to enhance an mRNA’s durability while adding a human β-globin UTR to the 3′-UTR has been demonstrated to enhance an mRNA’s stability and translational efficiency. Both the 3′-UTR and the 5′-UTR have been proven to protect mRNA against degradation. The effectiveness of mRNA vaccination can be increased by making nucleotide alterations to the 3′-UTR and 5′-UTR. Furthermore, Toll-like receptor activation may also occur via the use of intact IVT mRNA sequences (TLRs). The immunogenicity of IVT mRNA has been lowered by editing the ORF region with many changed nucleotides. Deliberately including pseudouridine has been proven to improve translation and lessen immunogenicity, for instance. N1-methyl-pseudouridine (m1)-modified messenger RNA also improved the translation efficiency. Recent studies have demonstrated that this increase is what causes the ribosome to recycle to the same mRNA from N1-methyl-enhanced pseudouridine. Aside from alterations to individual nucleotides, the immunogenicity of mRNA may also be decreased by codon optimization, based on GC-rich patterns. The mRNA ORF was created by altering the GC-rich codons utilized for each amino acid in a model of erythropoietin synthesis without chemical functionalization. These findings suggest that GC-enriched mRNA, despite causing excessive immunostimulation, may have substantial biological benefits [11,41,42].

## 6. Delivery Vehicles

### 6.1. Nanotechnology

The creation of innovative adjuvants and vaccination delivery methods has been greatly accelerated by rapid advancements in nanotechnology. The difficulty of guaranteeing the efficient distribution of mRNA vaccines and therapies is a major obstacle to realizing their full potential. Passive diffusion through the plasma membrane is impeded by nucleic acids because of their hydrophobicity and negative charge. Effective nucleic acid transport is further hampered by a number of hurdles, such as phagocyte absorption, interaction with serum proteins, and degradation by endogenous nucleases. Because of this, they require a method of delivery that can both ensure efficient cellular uptake and keep the drug stable during transport [43].

### 6.2. Lipid Nanoparticles

Lipid nanoparticles, with formulations containing cationic ionizable lipids, are utilized in the Moderna mRNA vaccine and Pfizer/BioNTech vaccine as the mRNA delivery vehicle. The most widely used nonviral nucleic acid carriers at the present time are lipid nanoparticles made from synthetic cationic lipids that interact at the nanoscale with polyanionic nucleic acids. Positively charged lipids help mRNA to self-assemble into nanoparticles, making it more stable and resistant to Rnase cleavage. After being endocytosed, the lipid nanoparticles convey their payload from the endosomes to the cytosol, where the antigenic proteins are produced from mRNA, encouraging the immune system to produce antibodies. In light of this, the basic phases of nucleic acid distribution are as follows: (i) lipid nanoparticle adsorption and endocytosis inside the cell, followed by (ii) nucleic acid release. Because biomembranes are frequently negatively charged, electrostatic interactions accelerate lipid nanoparticle fusion with the membrane, as well as adsorption. Unbinding the nucleic acid from a cationic lipid carrier is one of the crucial steps in nucleic acid delivery, whenever the nanoparticles enter the cell. It is thought to be the result of cellular anionic lipids neutralizing the cationic lipid charge, which causes nucleic acids to separate from nanoparticle carriers in two different ways: first, by neutralizing the cationic lipid charge and removing the electrostatic attraction between nucleic acids and lipids, and second, by disrupting the nanoparticle architecture and forming nonlamellar structures. The ability of cationic lipids to change the phase behavior of biomembrane lipids, and, more specifically, to encourage the development of nonlamellar lipid phases, has been hypothesized to be a factor in how successful they are as delivery agents. The phospholipid distearoylphosphatidylcholine (DSPC) is used in both the Moderna mRNA vaccine and BioNTech mRNA vaccine as a helper lipid in lipid nanoparticles. They contain an ionizable cationic lipid, DSPC, and cholesterol. As previously mentioned, because anionic nucleic acids create complexes, cationic lipids are essential parts of nonviral nucleic acid drug delivery systems. By preventing macrophage uptake, PEGylated lipids prolong the systemic circulation of the vehicles (“stealth” liposomes). Both phosphatidylcholines and cholesterol are biomembrane components that are widely used in lipid medicine delivery techniques. For the Pfizer/BioNTech vaccines, the PEG-lipid/cationic lipid/DSPC/cholesterol molar ratios are 1.6:46.3:9.4:42.7, and for Moderna vaccines, they are 50:1.5:38.5:10. There have been reports of particles of between 80 and 100 nm in size. Usually, 100 mRNA molecules are included in each lipid nanoparticle. SM-102 (Moderna) and ALC-0315 (Pfizer/BioNTech), the two cationic lipids used in this investigation, are both unique to the requisite businesses. It is important to note that the branched hydrocarbon chains found in the molecular structure of cationic lipids make them ideal for promoting nonlamellar phase formation, which is thought to increase the effectiveness of the lipid carrier’s delivery because it has a tight connection to membrane fusion and the consequent release of cargo. Other nanotechnology-based vaccine design techniques have been examined. One such technique resulted in the creation of nanoparticles that included the SARS-CoV-2 RBD, as well as numerous other unique RBDs from animal betacoronaviruses. These objects are mosaic nanoparticles created by protein conjugation using the SpyTag/SpyCatcher method. This method makes use of the 13-amino-acid peptide SpyTag, which produces an intermolecular peptide bond (12.3 kDa) when it spontaneously interacts with a synthetic protein domain, named SpyCatcher. Using this method, multimeric SARS-CoV-2 RBD nanoparticles were produced that strongly stimulated the production of neutralizing antibodies [44,45,46,47].

### 6.3. Self-Assembling Nanoparticles

Using ferritin nanoparticles that can self-assemble to show the S-protein of SARS-CoV-2 or its parts, other types of nanoparticle vaccinations were created. This type of immunization has been shown to produce cellular immunological responses and potent neutralizing antibodies in animal experiments. These nanoparticles are, therefore, thought to be a possible method of coronavirus vaccination against SARS-CoV-2 and other coronaviruses [48].

## 7. Imperative Aspects and Their Applications in COVID-19 Vaccines

### 7.1. Regulatory Scenario

This study uses WHO statistics and CAS-curated data to examine COVID-19 vaccine innovation. It compares vaccination platforms and top vaccines, based on clinical trial status. The importance of COVID-19’s defeat has boosted the development of vaccines and antibodies. More than 70 COVID-19 vaccines entered clinical testing a year following the disease’s outbreak. Several have provisional approval, and more may have this by 2021. While most vaccines in research employ conventional methods, mRNA vaccines and non-replicating adenovirus vaccines have won the battle for mass manufacture and distribution and provisional licensing. These accomplishments are the result of decades of scientific research, the timely dissemination of virus genome data, a collaborative approach between pharmaceutical organizations and academia, enhanced government aid, and indefatigable manpower efforts. Notwithstanding the significant progress made on the COVID-19 vaccination front, several concerns remain. Clinical study data show that the COVID-19 vaccines authorized thus far are successful at eliciting immunity, but it is unknown how long their protection will last. New research on adaptive immunity in COVID-19 patients revealed that eight months after the onset of symptoms, SARS-CoV-2 immunity may still be present. Despite the possibility of continued strong disease protection, a modeling study predicts that the neutralization titer will decline 250 days after inoculation with seven vaccines that have received provisional approval. More in-depth studies of the immunological reactions to vaccination are needed. Since its debut, COVID-19 immunizations have protected immunized people. In Europe, several agencies such as the Emergency Task Force (ETF), European Medicines Agency (EMA), European Medicines Regulatory Network (EMRN), European Union (EU), European Commission (EC), International Coalition of Medicines Regulatory Authorities (ICMRA), Medicines and Healthcare Products Regulatory Agency (MHRA), etc., have strongly supported and are supporting regulatory activities during the COVID-19 public health emergency. The latest data show that some immunizations may also prevent asymptomatic illness [49,50].

### 7.2. Mutations

Certain S-proteins of the SARS-CoV-2 variants are more infectious or immune to neutralizing antibodies. It is important to consider how mutations and variants affect vaccine efficacy. This topic has been researched extensively. In South Africa, where B.1.351 caused the most COVID-19 infections, Johnson & Johnson’s JNJ-78436735 vaccination was 57% effective. NVX-CoV2373 had a 48.6% overall efficiency against significant variants, with B.1.351 being the most common. The vaccine ADZ1222 is much less effective (10.4%) against B.1.351, according to a recent study. Pfizer/BioNTech and Moderna have created variation B.1.351-specific immunizations to be used as booster shots in clinical investigations. The COVID-19 vaccination pipeline comprises a variety of systems, which is encouraging [51].

### 7.3. Storage

Despite their effectiveness and ease of mass production, the two most popular mRNA vaccines must be maintained at specific temperatures. This may hinder the distribution of immunizations to less developed locations without specialized freezers. Vaccines based on other platforms that do not require precise preservation and shipping conditions are crucial. Since most leading immunizations require a booster shot, effective delivery mechanisms are needed to ensure that adequate booster shot regimens are followed. Long-lasting vaccines with one dose may have operational benefits and be more tolerated in specific locations [52].

## 8. COVID-19 Variants as Challenges to Vaccines

Coronaviruses have lower mutation rates than RNA viruses due to error-correcting enzymes. Proteogenomic studies of COVID-19 patients show a significant mutation rate. As in D614G, a spike-protein aspartic acid to glycine substitution, a mutation modifies the sequence. A variant, however, is a change in the genome’s sequence that manifests behavioral distinctions, such as transmissibility, virulence, and antigenicity. Since June 2020, the D614G variations have dominated globally. SARS-CoV-2 authorities in the UK identified VOC 202012/01, commonly known as B.1.1.7 or 20I/501Y.V1. This variation is more transmissible, without affecting hospitalization or case fatality. B.1.1.7 variations are 36–71% more transmissible than native variations. The loss at location 69/70del in VOC 202012/01 affects the S-gene diagnostic PCR tests. PCR techniques use many targets, so variations do not affect diagnosis. South Africa identified 501Y. V2 (B.1.351). The mutation has high transmissibility (over 50%) but there is no indication of clinical effects [53].

Versions in the UK and South Africa have the N501Y mutation; SARS CoV-2 N501Y could make the virus more contagious. This mutation threatens vaccine development. Using in silico FEP and REST, significantly more S1-RBD binding to STE90-C11 is seen in the N501Y mutant. STE90-C11 neutralizes COVID-19 in convalescent patients. STE90-C11 is also effective against RBD mutations. Sinopharm’s inactivated vaccine, Anhui Zhifei (an RBD-based protein subunit vaccine), and ZF2001 (a recombinant dimeric) neutralized human sera against the 501Y.V2 South African variants. The plaque reduction neutralization test (PRNT50) suggests the effectiveness of covaxin against the UK version of the virus. Several investigations have revealed that human convalescent serum cannot neutralize the SARS-CoV-2 E484K variants. BNT162b2 mRNA vaccination and convalescent vaccination-immunized sera-neutralized B.1.351 variations of SARS-CoV-2 were less effectively neutralized than B.1.1.248 variants, although there was coding for the N501 mutation and E484 mutation. Thus, mutations in S-protein NTD areas may contribute to viral resistance [54].

## 9. Vaccination: Protection

It is unknown whether vaccinated people spread the infection. The UK’s COVID-19 human challenge project will answer several questions. SARS-CoV-2 has 2.3 days to 3.3 days of doubling time and a reproduction number (R_0_) of 5.7. To cease transmission and create herd immunity, 82% of people must have immunity, either through immunization or prior illness. Increasing numbers of clinically important variations make herd immunity unlikely; therefore, SARS-CoV-2 mutations threaten vaccination efficacy. The evidence indicating that immunizations prevent severe COVID-19 sickness offers hope for the global population, which was halted by the pandemic. Despite these existing vaccines, the WHO says that physical distancing and masks are still the most effective way to reduce the infection burden. Statistics have predicted that after one month, a patient who reduced new infections in this way by 50% would enable a 27-fold decrease in the number of infections [55].

## 10. Vaccination Challenges

Vaccinations are being developed in response to the pandemic more quickly and in real time than ever before, yet there are still problems and difficulties. Most countries have poor vaccine coverage due to practical and social obstacles and the massive burden of world population vaccination. Differences in vaccination regulatory status between countries have hampered pandemic travel. This has increased vaccine reluctance and inequity because vaccines given in some countries are not recognized by others. Commercial competition, mass preordering, and the underreporting of data, instead of worldwide collaboration, characterize COVID-19 vaccine development. For fair access to safe and effective worldwide vaccinations, data reporting, global harmonization in vaccine development, production levels, distribution, and the suppression of commercial interests and social-political are needed. For herd immunity, countries need integrated surveillance and communication. COVAX was created by the WHO to ensure global vaccination distribution equity. The WHO, GAVI, and CEPI have pledged to distribute 2 billion doses by 2021. COVAX’s motto, “No one is safe unless everyone is safe”, emphasizes the idea of one safe planet [31].

By employing severe acute respiratory syndrome coronavirus 2 as a model, Zhu et al. created such a platform using the CRISPR engineering bacteriophage T4 to overcome various immunogenic obstacles. By inserting different viral components into the proper compartments of the phage nanoparticle structure, a pipeline of vaccine candidates was created. These comprise expressible spike genes in the genome, surface-decorated spike and envelope epitopes, and packed nucleocapsid proteins. In animal models, a phage ornamented with spike trimers was determined to be the most effective vaccination candidate. This vaccine induced strong immune responses in both TH1 and TH2 immunoglobulin G subclasses, inhibited virus-receptor interactions, neutralized viral infection, and provided complete protection against viral challenges without the need for an adjuvant [56,57].

## 11. Conclusions

In conclusion, despite numerous unanswered questions and many challenges, the incredible developments offer reassurance that this pandemic may be stopped short because of vaccine development. According to the reports, a large number of pharmaceutical companies are working to produce a COVID-19 vaccine. In several nations throughout the world, many of these vaccines have received emergency use approval, and some are now undergoing various stages of clinical research. A newly evolved form of COVID-19, which poses a new hazard, has recently been reported in several parts of the world. The fact that immunizations can effectively stop serious illness is a glimmer of optimism. Additionally, the vaccines that are now in development are being tested against these altered versions and will be released in the near future. However, until then, people must practice social restraint, maintain physical separation, and wear protective masks.

## Figures and Tables

**Figure 1 vaccines-10-01689-f001:**
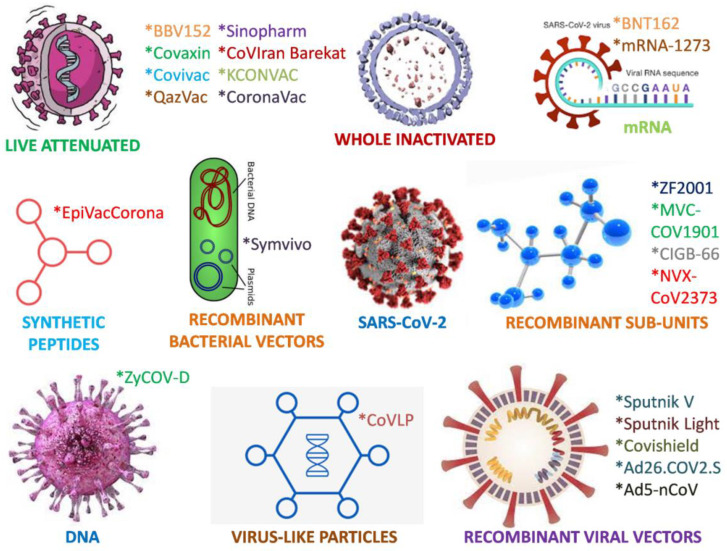
Various types of coronavirus vaccines and their examples.

**Figure 2 vaccines-10-01689-f002:**
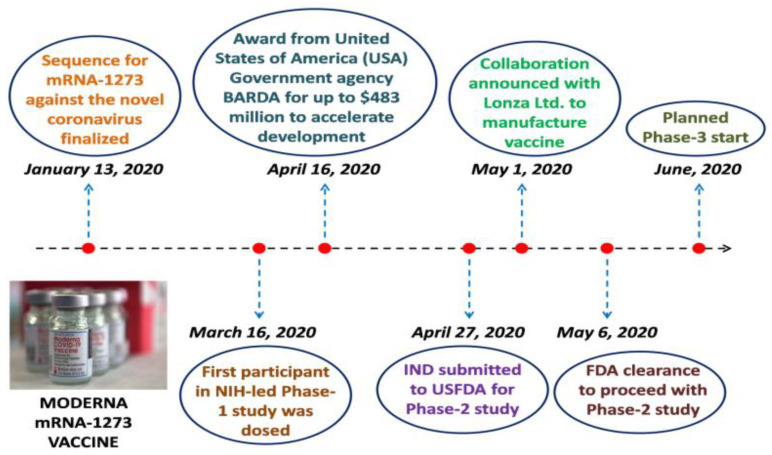
The developmental progress of the Moderna mRNA-1273 vaccine.

**Table 1 vaccines-10-01689-t001:** Advanced vaccine candidates.

Vaccine Platform	Organization	Candidate	Immune Response	Country Involved	Key Trials
RNA-based vaccine	BioNTech/Pfizer	BNT162(3LNP-mRNAs)	Spike protein; S-2P (full-length with proline substitutions, K986P and V987P)	Germany/United States	Phase 3 NCT04368728
	Moderna/National Institute of Allergy and Infectious Diseases	mRNA-1273	Spike protein; S-2P (full-length with proline substitutions, K986P and V987P)	United States	Phase 3 NCT04470427
	CureVac	CVnCoV Vaccine	Prefusion stabilized full-length spike protein	Germany	Phase 3 NCT04674189
DNA-based vaccine	Inovio Pharmaceuticals + International Vaccine Institute + Advaccine Biopharmaceuticals	INO-4800 + Electroporation	Full-length spike protein	United States	Phase 2/3 NCT04642638
	AnGes + Takara Bio + Osaka University	AG0301-COVID19	Spike protein	Japan	Phase2/3 NCT04655625
	ZydusCadila	nCov vaccine (ZyCoV-D)	Spike protein	India	Phase 1/2 CTRI/2020/07/026352
Non-Replicating Viral Vector	CanSino	Novel coronavirus vaccine(adenovirus type 5 vector)	Full-length spike protein	China	Phase 3 NCT04526990
	Gamaleya Research Institute	Sputnik V Gam-COVID-Vacadeno-based (rAd26-S + rAd5-S)	Full-length spike protein	Russia	Phase 3 NCT04530396
	Janssen	Ad26.COV2.S	Full-length S with two proline substitutions (K986P and V987P) and two mutations at the furin cleavage site (R682S and R685G)	United States	Phase 3 NCT04505722
	University of Oxford/AstraZeneca	ChAdOx1-S-(AZD1222)	Full-length spike protein	UK	Phase 3 NCT04516746
Replicating Viral Vectors	Jiangsu Provincial Center for Disease Prevention and Control	DelNS1-2019-nCoV-RBD-OPT1	Spike protein	China	Phase 2 ChiCTR2000039715
Protein Subunit	Anhui Zhifei	Recombinant SARS-CoV-2 vaccine (CHO cell)	RBD dimer (as tandem repeat residues 319–537)	China	Phase 3 NCT04445194
	CIGB	CIGB 66	RBD + aluminum hydroxide	Cuba	Phase 1/2 RPCEC00000345
	Medigen	MVC-COV1901	Spike protein with aluminum hydroxide and CpG1018	Taiwan	Phase 2 NCT04695652
	Novavax	SARS-CoV-2 rS (CHO)/Matrix M1 adjuvant (NVX-CoV2373)	Full-length spike protein with two proline substitutions(K986P and V987P) and threemutations at cleavage site (R682Q, R683Q, R685Q)	United States	Phase 3 NCT04611802
	Clover Biopharm/GSK	SCB 2019 + AS03 or CpG 1018 adjuvant plus alum adjuvant	Ectodomain of wild-type S with fusion to trimer-tag	Australia	Phase 2/3 NCT04672395
	Covaxx + United Biomedical Inc	UB 162	Multitope S1-RBD peptide based	China	Phase 2/3 NCT04683224
Live Attenuated Virus	Codagenix/Serum Institute India	COVI-VAC	Whole virus	India	Phase 1 NCT04619628
Inactivated Virus	Sinopharm/Beijing Institute of Biological Products/Wuhan Institute of Biological Products	SARS-CoV-2 vaccine	Whole virus	China	Phase 3 ChiCTR2000034780
	Bharat Biotech	Whole virion inactivated SARS-CoV-2 vaccine (BBV152)	Whole virus	India	Phase 3 NCT04641481
	Institute of Medical Biology/Chinese Academy of Medical Sciences	SARS-CoV-2 vaccine	Whole virus	China	Phase 3 NCT04659239
	Sinovac	CoronaVac	Whole virus	China	Phase 3 NCT04456595
	Chumakov Center	CoviVac	Whole virus	Russia	-
	Shifa Pharmed	COVIranBarekat	Whole virus	Iran	Phase 2/3 IRCT20201202049567N3
	Minhai Co	KCONVAC	Whole virus	China	Phase 3 NCT04852705
	Research Institute for Biological Safety Problems	QazCovid-in	Whole virus	Kazakhstan	Phase 3 NCT04530357
Virus-like Particles	Medicago/GSK	Coronavirus-like particle COVID-19 (CoVLP) with AS03 adjuvant	Living plant-based platform to produce noninfectious VLP	Canada	Phase 2/3 NCT04636697
Bacterial vector	Symvivo	bacTRL-Spike Vaccine	Spike protein	Canada	Phase 1 NCT04334980
Synthetic peptide	Vektor State Research Center of Virology and Biotechnology	EpiVacCorona	Spike protein	Russia	Phase1/2 NCT04527575

## Data Availability

As this is a review article, there was no data collected from patients.

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
