# Peer review of "Comprehensive Highlights of the Universal Efforts towards the Development of COVID-19 Vaccine"

_vaccines, 2022, doi:10.3390/vaccines10101689_

Round 1
Reviewer 1 Report
The review is well written and an important contribution to the field of COVID-19. The authors cover a range of vaccines and vaccine delivery platforms for COVID19 vaccines.
Some corrections are requested.
1. Line 104 & 105: consider rephrasing ".. This research.." to "This review."
2. Please consider adding panel labels in figure 1 and mention the name of different coronavirus vaccines in the legend.
It would be nice to add a section describing intranasal vaccine delivery platforms with a focus on Phage based vaccines as these are emerging as new technologies for COVID-19 vaccines.
Zhu J, Ananthaswamy N, Jain S, Batra H, Tang WC, Lewry DA, Richards ML, David SA, Kilgore PB, Sha J, Drelich A, Tseng CK, Chopra AK, Rao VB. A universal bacteriophage T4 nanoparticle platform to design multiplex SARS-CoV-2 vaccine candidates by CRISPR engineering. Sci Adv. 2021 Sep 10;7(37):eabh1547. doi: 10.1126/sciadv.abh1547. Epub 2021 Sep 8. PMID: 34516878; PMCID: PMC8442874.
Batra H, Zhu J, Jain S, Ananthaswamy N, Mahalingam M, Tao P, Lange C, Zhong C, Kearney M.F, Hu H, Maldarelli F, Rao VB. Engineered Bacteriophage T4 Nanoparticle as a Potential Targeted Activator of HIV-1 Latency in CD4+ Human T-cells. bioRxiv 2021.07.20.453091; doi: https://doi.org/10.1101/2021.07.20.453091
Author Response
Point 1: Line 104 & 105: consider rephrasing ".. This research.." to "This review."
Response-1: Thank you very much for the comment. We have now rephrased ".. This research.." to "This review.".
Point 2: Please consider adding panel labels in figure 1 and mention the name of different coronavirus vaccines in the legend.
Response-2: Thank you very much for the comment. We have mentioned the name of different coronavirus vaccines in the legend of Figure 1.

Reviewer 2 Report
Dear Authors,
I read this work with interest due to the topic being pointed out in the title and abstract. I am finding this kind of review really needed and interesting, trying to update and summarize in this case, how the vaccine development is currently. I appreciate the effort of writing a paper like this.
The introduction is well constructed and written. A clear description of the pandemic emergency and the need for a quick successful vaccine.
Through the following paragraphs, numbered 2 and 3, start a quick description of all different kinds of vaccines. Some descriptions are a bit shot and shallow, but mainly lacking of references. For example, block 3.1., but others also, like 3.3.1 lines 280-290 or whole 3.6.
I found weak the information contained in paragraph 7.1 and not much connected with its following ones, 7.2 and 7.3. A kind of miscellaneous part in the middle of the paper.
I enjoyed reading the information contained in paragraphs 5, 6 and 8 because although short, the authors summarized well some concepts and current information useful for the reader of a review like this. Here was possible to extract some interesting conclusions from the different vaccine strategies.
In general, table 1 is the key to the manuscript, but the surrounding text is not deep and structured enough as a review should be, with the exception of the last mentioned paragraphs.
Author Response
Point 1: Through the following paragraphs, numbered 2 and 3, start a quick description of all different kinds of vaccines. Some descriptions are a bit shot and shallow, but mainly lacking of references. For example, block 3.1., but others also, like 3.3.1 lines 280-290 or whole 3.6.
Response-1: Thank you very much for the comment. All bibliography provided for the support of section 3 is now separately mentioned in the first paragraph.
Point 2: I found weak the information contained in paragraph 7.1 and not much connected with its following ones, 7.2 and 7.3. A kind of miscellaneous part in the middle of the paper.
Response-2: Thank you very much for the comment. We have now changed the title of the section as “IMPERATIVE ASPECTS AND THEIR APPLICATIONS IN COVID-19 VACCINES” to match this Section with the rest of the manuscript.
Point 3: I enjoyed reading the information contained in paragraphs 5, 6 and 8 because although short, the authors summarized well some concepts and current information useful for the reader of a review like this. Here was possible to extract some interesting conclusions from the different vaccine strategies.
Response-3: Thank you very much for the comment. We have improved the Conclusion section accordingly.
Point 4: In general, table 1 is the key to the manuscript, but the surrounding text is not deep and structured enough as a review should be, with the exception of the last mentioned paragraphs.
Response-4: Thank you very much for the comment. We have improved the surrounding text accordingly.

Reviewer 3 Report
This review from Riyaz Ahamed Shaik et al., is a valuable resource as the authors themselves state at the end of the introduction. I found this work very useful for any interested reader that would focus on different aspects of COVID-19 vaccine development and it collects a very exhaustive and complete state of the art of the COVID-19 vaccines until now developed and also under investigation through specific trials. The abstract conveys the main points of the manuscript and its reading appear fluid and scientifically sound, as well as the entire manuscript. I only would suggest the authors a thorough check of the writing before final publication because some minor typos need to be fixed along with some missing information that need to be integrated in the final version of the manuscript. I have picked up some below.
- I would suggest to change “Toward” in “Towards” in the title
- Line 47: the authors should complete the sentence specifying “a few examples” of what they are referring to. In other words, they should explicitly refer to COVID-19 proactive measures or at least introduce briefly, with few words, the COVID-19 context. This latter is introduced after from line 48, but I would suggest them to also introduce it before in the first paragraph of the introduction.
- Line 51: remove “the” before “economy”:
- Line 53: Please substitute "COVID vaccine candidates" with "COVID-19 vaccine candidates".
- From line 56 to 62 a dedicated bibliography about vaccination and typical examples of pathogens eradication through vaccine administration are missing. Please insert it appropriately.
- From line 66 to 80, a mention about in silico tools or computational approaches in speeding up the COVID-19 vaccine development process is recommended. I would suggest the authors to include also some computational examples and benefits they provided and they are going to provide in vaccine discovery and development pipeline. Please see some dedicated literature such as the following ones: 1) https://doi.org/10.1016/j.imu.2022.100862; 2) https://doi.org/10.1186/s12859-020-03872-0 or feel free to insert many others that literature is offering.
- From line 81-84, the authors should also mention the correspondent rating groups present in Europe that supported and are supporting regulatory activities during public-health emergency like COVID-19 (for example COVID-19 EMA Pandemic Task Force and equivalent ones).
- Line 86: Please change "COVID vaccine candidates" with "COVID-19 vaccine candidates".
- From line 116 to 126, for the sake of clarity, I would suggest to include also a mention of the Cuban vaccine against COVID-19.
- In line 161, a misprint is present: “SARS-S CoV-2 protein”. Please fix it.
- The authors should insert a proper bibliography in section 3 and also in each subsections, and eventually include the clinical trial Identifier or sponsor. Indeed, in such part of section 3 specific references are missing. I would recommend the authors to check carefully this fundamental section of the review.
- In Figure 1, please revise “RECOMBINANT SUB-UNITS” in “RECOMBINANT SUBUNITS”.
- In line 675, a full stop is missing.
Author Response
Point 1: I would suggest to change “Toward” in “Towards” in the title.
Response-1: Thank you very much for the comment. We have changed “Toward” in “Towards” in the title.
Point 2: Line 47: the authors should complete the sentence specifying “a few examples” of what they are referring to. In other words, they should explicitly refer to COVID-19 proactive measures or at least introduce briefly, with few words, the COVID-19 context. This latter is introduced after from line 48, but I would suggest them to also introduce it before in the first paragraph of the introduction.
Response-2: Thank you very much for the comment. We have added a paragraph in the introduction where COVID-19 proactive measures are mentioned.
Point 3: Line 51: remove “the” before “economy”.
Response-3: Thank you very much for the comment. We have removed “the” before “economy”.
Point 4: Line 53: Please substitute "COVID vaccine candidates" with "COVID-19 vaccine candidates".
Response-4: Thank you very much for the comment. We have substituted "COVID vaccine candidates" with "COVID-19 vaccine candidates".
Point 5: From line 56 to 62 a dedicated bibliography about vaccination and typical examples of pathogens eradication through vaccine administration are missing. Please insert it appropriately.
Response-5: Thank you very much for the comment. We have mentioned the typical examples of pathogens eradication through vaccine administration.
Point 6: From line 66 to 80, a mention about in silico tools or computational approaches in speeding up the COVID-19 vaccine development process is recommended. I would suggest the authors to include also some computational examples and benefits they provided and they are going to provide in vaccine discovery and development pipeline. Please see some dedicated literature such as the following ones: 1) https://doi.org/10.1016/j.imu.2022.100862; 2) https://doi.org/10.1186/s12859-020-03872-0 or feel free to insert many others that literature is offering.
Response-6: Thank you very much for the comment. We have added computational role in COVID-19 vaccine development. The above-mentioned references are also cited.
Point 7: From line 81-84, the authors should also mention the correspondent rating groups present in Europe that supported and are supporting regulatory activities during public-health emergency like COVID-19 (for example COVID-19 EMA Pandemic Task Force and equivalent ones).
Response-7: Thank you very much for the comment. We have provided the European Agencies that supported and are supporting regulatory activities in COVID-19 emergency.
Point 8: Line 86: Please change "COVID vaccine candidates" with "COVID-19 vaccine candidates".
Response-8: Thank you very much for the comment. We have changed "COVID vaccine candidates" with "COVID-19 vaccine candidates".
Point 9: From line 116 to 126, for the sake of clarity, I would suggest to include also a mention of the Cuban vaccine against COVID-19.
Response-9: Thank you very much for the comment. We have now mentioned Cuban COVID-19 vaccine in the manuscript.
Point 10: In line 161, a misprint is present: “SARS-S CoV-2 protein”. Please fix it.
Response-10: Thank you very much for the comment. We have now corrected the “SARS-S CoV-2 protein” with “SARS-CoV-2”.
Point 11: The authors should insert a proper bibliography in section 3 and also in each subsections, and eventually include the clinical trial Identifier or sponsor. Indeed, in such part of section 3 specific references are missing. I would recommend the authors to check carefully this fundamental section of the review.
Response-11: Thank you very much for the comment. All bibliography provided for the support of section 3 is now separately mentioned in the first paragraph.
Point 12: In Figure 1, please revise “RECOMBINANT SUB-UNITS” in “RECOMBINANT SUBUNITS”.
Response-12: Thank you very much for the comment. We have revised “RECOMBINANT SUB-UNITS” in “RECOMBINANT SUBUNITS”.
Point 13: In line 675, a full stop is missing.
Response-13: Thank you very much for the comment. We have now provided the full stop.

Round 2
Reviewer 2 Report
Dear Authors,
Thanks for considering and the comments and try to improve the manuscript. Due to the interest of the topic it deals with, I am finding that this review is ready to be published